# Mitochondrial Unfolded Protein Response and Integrated Stress Response as Promising Therapeutic Targets for Mitochondrial Diseases

**DOI:** 10.3390/cells12010020

**Published:** 2022-12-21

**Authors:** Hedong Lu, Xiaolei Wang, Min Li, Dongmei Ji, Dan Liang, Chunmei Liang, Yajing Liu, Zhiguo Zhang, Yunxia Cao, Weiwei Zou

**Affiliations:** 1Reproductive Medicine Center, Department of Obstetrics and Gynecology, The First Affiliated Hospital of Anhui Medical University, No. 218 Jixi Road, Hefei 230022, China; 2NHC Key Laboratory of Study on Abnormal Gametes and Reproductive Tract, Anhui Medical University, No. 81 Meishan Road, Hefei 230032, China; 3Key Laboratory of Population Health Across Life Cycle, Anhui Medical University, Ministry of Education of the People’s Republic of China, No. 81 Meishan Road, Hefei 230032, China; 4Anhui Province Key Laboratory of Reproductive Health and Genetics, No. 81 Meishan Road, Hefei 230032, China; 5Biopreservation and Artificial Organs, Anhui Provincial Engineering Research Center, Anhui Medical University, No. 81 Meishan Road, Hefei 230032, China

**Keywords:** mitochondrial unfolded protein response, integrated stress response, mitochondrial diseases, mitochondrial function

## Abstract

The development and application of high-throughput omics technologies have enabled a more in-depth understanding of mitochondrial biosynthesis metabolism and the pathogenesis of mitochondrial diseases. In accordance with this, a host of new treatments for mitochondrial disease are emerging. As an essential pathway in maintaining mitochondrial proteostasis, the mitochondrial unfolded protein response (UPR^mt^) is not only of considerable significance for mitochondrial substance metabolism but also plays a fundamental role in the development of mitochondrial diseases. Furthermore, in mammals, the integrated stress response (ISR) and UPR^mt^ are strongly coupled, functioning together to maintain mitochondrial function. Therefore, ISR and UPR^mt^ show great application prospects in the treatment of mitochondrial diseases. In this review, we provide an overview of the molecular mechanisms of ISR and UPR^mt^ and focus on them as potential targets for mitochondrial disease therapy.

## 1. Introduction

Mitochondria are indispensable for the survival of the majority of eukaryotes. Mitochondria have an independent mitochondrial genome (mtDNA) that, in humans, encodes 13 proteins related to oxidative phosphorylation (OXPHOS). These proteins are all enzyme subunits located in the electron transport chain (ETC). Other ETC enzyme subunits are encoded by the nuclear genome (nDNA) and then imported into the mitochondria [1]. Thus, rather than being independent, mitochondria are closely associated with the nucleus. Furthermore, with the deepening of the understanding of mitochondria, recent research indicates that mitochondria play an important role in the metabolism of bioactive substances required by cells, such as fatty acid oxidation, heme metabolism, and one-carbon metabolism in addition to their vital role in bioenergetics [2]. The UPR^mt^ is a pathway for retrograde signaling from mitochondria to the nucleus that is used to preserve mitochondrial proteostasis [3,4].

When protein-folding errors occur in the mitochondria, the UPR^mt^ functions in the maintenance of protein homeostasis by activating mitochondrial chaperone proteins and proteases to help with the correct folding of proteins or the removal of misfolded proteins [5]. The UPR^mt^ also protects other aspects of mitochondrial function, such as in preventing electron transport chain dysfunction by promoting the assembly of oxidative-phosphorylation-related subunits and the synthesis of coenzyme Q, for example [6]. Mitochondrial diseases constitute a group of genetic disorders characterized by oxidative phosphorylation defects leading to impaired ATP synthesis [7,8,9]. We therefore raise the question of whether the UPR^mt^ can be targeted as a potential therapeutic approach in treating mitochondrial diseases.

In mammals, there is increasing evidence that the mitochondrial integrated stress response (ISR) is closely linked with the UPR^mt^ [10]. By phosphorylating serine residues in the α-subunit of eIF-2, the ISR can prompt a decrease in total protein translation and an increase in the synthesis of proteins involved in environmental adaptation [11]. In fact, translation attenuation is emerging as a potential method for treating mitochondrial diseases [12].

With further insight into the mechanisms of UPR^mt^ and ISR, their functions and interactions in mitochondrial diseases have gradually attracted attention. In this review, we focus on the molecular mechanisms and interconnection of UPR^mt^ and ISR as well as their roles in the pathogenesis of mitochondrial diseases. We also discuss their possible applications in the therapy of mitochondrial diseases.

## 2. UPR^mt^

### 2.1. In C. Elegans

The UPR^mt^ is a stress response in which mitochondria initiate the transcriptional activation of a set of genes encoded by nuclear DNA, such as for mitochondrial heat shock proteins and proteases, to maintain protein homeostasis [4,13]. Some key activators required for this response have been identified, including CLPP, UBL-5, DVE-1, ATFS-1, LONP-1, and HAF-1 [6,14]. The corresponding pathway has also been described: unfolded proteins are degraded into polypeptides by the CLPP protease in the mitochondrial matrix, and the polypeptides are transported into the cytoplasm via the HAF-1 matrix polypeptide transporter to activate the UBL-5 and DVE-1 transcriptional complex factors [15] (Figure 1). However, this needs to be verified through more research.

The activation of the UPR^mt^ also requires chromatin reorganization, which is dependent on the MET-2 and LIN-65 histone methyltransferases. Normally, LIN65 is located in the cytoplasm, while DVE-1 and UBL-5 are distributed around the perinuclear region. MET-2 is activated to promote the redistribution of LIN65 to the perinuclear region during stress conditions. Then, LIN65 facilitates DVE-1 and UBL-5 redistribution and chromatin recombination to the promoters of UPR^mt^-responsive genes [16,17] (Figure 1).

As a critical regulator of the UPR^mt^, ATFS-1 possesses a nuclear localization sequence, a C-terminal leucine zipper domain, and an N-terminal mitochondrial-targeting sequence. Under normal conditions, ATFS-1 is transported to the mitochondrial matrix for degradation by the LONP-1 protease [18]. However, when a number of triggers lead to impaired ATFS-1 transportation, such as homeostasis disturbance, hypoxia, and abnormal mitochondrial membrane potential, ATFS-1 accumulates around the nucleus to initiate a nuclear-regulated mitochondrial repair program [6] (Figure 1). Although ATFS-1 accumulation leads to the reduced transcription of OXPHOS-related subunits, coding genes, and tricarboxylic acid cycle (TCA)-related genes during mitochondrial dysfunction, more than 50 genes related to mitochondrial ribosome function are upregulated, as are genes required for mtDNA replication and genes of the cardiolipin biosynthesis pathway, which are essential for mitochondrial inner membrane synthesis. Moreover, genes required for the import of mitochondrial proteins and OXPHOS complex assembly are also upregulated. In addition, the expression of more than 20 genes involved in reactive oxygen species (ROS) clearance is induced to in an ATFS-1-dependent manner. Overall, mitochondrial function is ultimately restored as a result of these biological changes [19,20]. Although the regulatory mechanism of the UPR^mt^ in mammals is similar to that of *C. elegans*, it also has its distinct characteristics.

### 2.2. In Mammals

Activation of the UPR^mt^ in mammals is mediated by the transcription factors CHOP, ATF4, and ATF5, which is a homolog of ATFS-1 [21,22]. As research has progressed, there is mounting evidence that the ISR behaves in a manner highly consistent with the UPR^mt^ in mammals [23]. The ISR-promoted phosphorylation of eukaryotic translation initiation factor 2 (eIF2α) stimulates the transcription of CHOP, ATF4, and ATF5, which contribute to activation of the UPR^mt^ [24,25], whereas the ISR is not required for activation of the UPR^mt^ in *C. elegans* [26]. In addition, additional proteins involved in the UPR^mt^, such as the entire sirtuin family, were identified in a recent study [27]. The sirtuin family is a group of nicotinamide adenine dinucleotide (NAD^+^)-dependent deacetylases, and each of its seven homologs (SIRT1–7) in mammals performs a different biological function. Among them, SIRT1 is closely related to the UPR^mt^. UPR^mt^ activators, such as heat shock protein-60 (Hsp60, a molecular chaperone in the mitochondria) and CLPP, are expressed in response to increased SIRT1 activity. This is consistent with the findings in the skeletal muscles of respiratory-chain-deficient mice. That is, when SIRT1 activity is induced by NAD^+^ supplementation, there is a simultaneously enhancement in the expression levels of Hsp60 and CLPP, which are initially low due to respiratory chain defects [28]. Similarly, increased SIRT1 activity due to NAD^+^ supplementation results in UPR^mt^ activation and tissue regeneration [29]. In addition, SIRT3 may also play an integral role in the UPR^mt^. Studies on primary hepatocytes have shown that Hsp60 and CLPP expression are significantly downregulated in the absence of SIRT3 and then increase after NAD^+^ supplementation [30]. In contrast, SIRT7 usually inhibits UPR^mt^ activation. SIRT7 targets the promoters of mitochondrial ribosomal proteins (mRPs) and mitochondrial translation factors (mTFs) under the action of nuclear respiratory factor 1 (NRF1), resulting in the suppression of transcription levels and, thus, a reduction in mitochondrial protein levels [31]. The molecular mechanism of the UPR^mt^ deserves continued exploration, as it may be an effective way to treat mitochondrial diseases.

## 3. ISR

The integrated stress response is a signal pathway activated in cells in response to various stimulation and physiological changes [32]. The ISR regulates protein translation by managing the concentration of the ternary complex (TC). The TC is composed of eIF2 (consisting of α, β, and γ subunits), guanosine 5′-triphosphate (GTP), and charged methionyl-initiator tRNA (Met-tRNAi). The TC is essential for the initiation of the AUG transcription initiation factor [33]. Under the control of four different kinases, namely GCN2, PERK, HRI, and PKR, phosphorylation of the serine residues of the eIF2α subunit is stimulated, which influences the TC concentration, and the overall level of protein synthesis is reduced while there is increased synthesis of stress-related proteins involved in adaptation to environmental changes [10,11,34] (Figure 2A). For example, when mitochondrial function is impaired, eIF2α phosphorylation increases the transcription of CHOP, ATF4, and ATF5, causing a series of responses toward restoring mitochondrial function [35,36,37]. In addition to the four specialized eIF2 kinases that phosphorylate eIF2, there are two specific phosphatases that antagonize this reaction. Both phosphatases contain a common catalytic core subunit (protein phosphatase 1 (PP1)) and a regulatory subunit (GADD34 or CReP) that allow the phosphatase to act specifically on eIF2 [38].

Activation of the ISR during mitochondrial respiratory chain dysfunction is mainly achieved through two kinases: HRI and GCN2 (Figure 2B). For the first, mitochondria dysfunction activates OMA1 (a protease localized to the inner mitochondrial membrane) to promote the conversion of DELEL to DELES (a protein involved in the ISR that has two forms). The latter activates HRI, which then leads to an increase in ATF4 expression [12,39]. For the second, mitochondrial dysfunction reduces cytoplasmic aspartate and asparagine, leading to GCN2 activation, which in turn activates the ISR [40,41] (Figure 2B). After ISR activation, the downstream molecules of uORFS, CHOP, ATF4, and ATF5 are preferentially translated. Normally, uORFS inhibits downstream translation. Nevertheless, eIF2α phosphorylation resulting from ISR activation inhibits the action of uORFS, thereby relatively increasing its downstream translation [42].

CHOP expression is increased in mitochondrial dysfunction and interacts with C/EBPβ, which tends to accumulate in the perinuclear region, and then regulates ATF4. ATF4 plays an important role in the UPR^mt^ [43,44]. As an important downstream target gene of the ISR, ATF4 can stimulate the alteration of cellular metabolites, especially serine. Serine contributes to the pathogenesis of mitochondrial diseases, including, for instance, mitochondrial-related alcoholic liver disease [21,45]. In mammals, ATF5 is a crucial regulator of the UPR^mt^. A recent study of point mutations causing mitochondrial diseases found that knockdown of ATF5 inhibits excessive activation of the UPR^mt^, resulting in improved mitochondrial function [22,46].

## 4. Relationship between UPR^mt^ and ISR

Both the UPR^mt^ and ISR are activated during mitochondrial dysfunction and take part in the pathogenesis of mitochondrial diseases [47,48,49]. In mammals, the important role of the ISR in UPR^mt^ activation is gradually being explored [23,50]. The ISR promotes UPR^mt^ activation, and the UPR^mt^ leads to a mild increase in eIF2α phosphorylation, which can reduce the stress on chaperone proteins and proteases. All of this contributes to the maintenance of mitochondrial function. Moderate activation of the ISR and UPR^mt^ can counteract stress and improve mitochondrial function. For example, in *C. elegans*, UPR^mt^ activation can enhance OXPHOS assembly, protein homeostasis, and coenzyme Q synthesis and inhibit the transcription of OXPHOS subunits. The overall effect of these responses is to promote mitochondrial repair, which extends the lifespan of *C. elegans* with mild mitochondrial dysfunction [21,51,52]. Activation of the ISR reduces total protein translation and promotes serine biosynthesis, one-carbon metabolism, and proline biosynthesis, resulting in adaptive metabolic changes. For example, the inhibition of protein synthesis by rapamycin has extended the lifespan of complex-I-deficient mice [53,54].

## 5. Mitochondrial Diseases

Mitochondrial diseases (MDs) are a group of inherited defects in oxidative phosphorylation leading to impaired ATP synthesis and insufficient energy [55,56]. The prevalence of MDs is as high as 1:5000. The clinical manifestations are heterogeneous, with single-system or multisystem involvement. The age of onset ranges from newborns to the elderly, and it can progress to the point of severe clinical symptoms. Organs with high energy requirements are the most commonly affected, such as the skeletal muscles, nervous system, etc. [57,58,59,60,61]. Because nuclear DNA (nDNA) and mitochondrial DNA (mtDNA) are both involved in the encoding of complexes in the electron transport chain, mitochondrial diseases can be triggered by mutations in either nDNA or mtDNA [62]. The pathogeneses of MDs are diverse. The same mutation of mtDNA can cause different phenotypes, whereas different mutations of mtDNA or nDNA can cause the same phenotype. Moreover, a particular phenotype is often caused by a combination of multiple mutations [63,64].

### 5.1. MDs Caused by Mutations in mtDNA

In humans, mtDNA encodes 13 structural proteins of the mitochondrial respiratory chain, and the rest are encoded by nDNA [65]. The reason for the high mutation rate of mtDNA is that mtDNA is not protected by histones and chromatin and has low mitochondrial repair enzyme activity [66,67]. Nevertheless, cells can tolerate mtDNA mutations until they exceeds a certain threshold rate (such as a 70% mutation rate) and become symptomatic [68]. There are two mutational patterns in mtDNA that result in the development of MDs. The first is point mutation, such as MELAS, which is usually inherited from the mother and is clinically characterized by recurrent shock, myopathy, ataxia, myasthenia, mental retardation, and deafness beginning before the age of forty [69]. The second is large deletion, which commonly occurs during embryonic development, such as in the case of CPEO, which is clinically characterized by chronic, bilateral progressive ptosis; the gradual onset of ocular motility disorders; impaired eye movement in all directions; and eventual fixation of the eyes [70] (Table 1). Some nDNA mutations can also lead to secondary mutations in mtDNA, usually large deletions and point mutations that thus lead to MDs [71].

### 5.2. MDs Caused by Mutations in nDNA

Unlike mtDNA, which is inherited maternally, nDNA follows classical Mendelian inheritance. Since the first case of MD caused by a nDNA mutation was reported in 1985 [72], various nDNA mutations causing MDs have been successively identified: first, mutations in genes encoding respiratory chain subunits, such as those associated with complex I activity [73,74]; second, nuclear genes encoding proteins involved in mtDNA replication or transcription, such as the TP gene associated with MNGIE [75,76]; and third, mutations in genes relating to protein assembly and the maintenance of protein function, such as hereditary spastic paraplegia [77] (Table 1). With the advent of next-generation sequencing (NGS) technology, the discovery of pathogenic mutations has advanced at a breathtaking pace, and more than 300 pathogenic mutations affecting OXPHOS have been identified to date [78,79]. Establishing a genetic diagnosis is a challenging process due to the lack of correlation between clinical phenotypes and genotypes, variability in clinically affected tissues, and diversity of mutation patterns. The study of these mutations not only helps to elucidate the pathogenesis of MDs, but also provides evidence for the diagnosis of MDs and potential therapeutic directions [80,81]. Thus far, pathogenic mutations in nDNAs are more prevalent in pediatric cases, while mtDNA mutations are more common in adults [7].

## 6. Roles of UPR^mt^ and ISR in Mitochondrial Diseases

The mechanism underlying the development of mitochondrial diseases is respiratory chain dysfunction due to various causes. The improvement and application of high-throughput omics technologies have enabled a more complete understanding of the physiological function of mitochondria (a detailed and understandable overview can be found in [86]). In the event of mitochondrial dysfunction, activated UPR^mt^ increases the production of ATP by improving glucose uptake, enhancing glycolysis, promoting the assembly of OXPHOS-related subunits, and improving mitochondrial antioxidant capacity by regulating the related genes. The UPR^mt^ effectively reduces the mitochondrial load and repairs mitochondrial function through the abovementioned ways. If the UPR^mt^ fails to reduce the mitochondrial load, mitophagy is initiated via the mitophagy–lysosome pathway. The dominant intracellular mitophagy pathway is mediated by PINK1 (PTEN-induced putative kinase 1) and Parkin (E3 ubiquitin ligases). The timely and appropriate removal of damaged or aged mitochondria through selective mitophagy is important for maintaining intracellular homeostasis. A recent study found that the activation of mitophagy alleviated mitochondrial dysfunction in the skeletal muscles of aged mice and human patients [87]. In another study, ferroptosis was found to be associated with OXPHOS deficiency and mitochondrial disease, and Oma1-Dele1-mediated ISR was shown to have antagonistic effects on ferroptosis in mitochondrial cardiomyopathy cardiomyocytes [88]. Similarly, the ISR protects cells by altering the expression of related substances, such as nucleotides, serine, and the FGF21 and GDF15 cellular metabolic factors. However, if stress is not reduced, the ISR triggers apoptosis to eliminate damaged cells. This suggests that activation of the ISR and UPR^mt^, at least in some mitochondrial diseases, can have a protective effect. The results from applications of metabolomics, genomics, and proteomics in other mammalian and human MD models also support this theory.

Although activation of the UPR^mt^ helps to promote the recovery of mitochondrial function, some studies have also shown that UPR^mt^ overactivation leads to some detrimental outcomes, such as increased accumulation of mutant mtDNA and exacerbation of mitochondrial damage. Moreover, in mammals, the mechanism of UPR^mt^ action in mitochondrial function may be more complex and needs to be further investigated [46,89,90]. The ISR is inseparable from the UPR^mt^, and studies have also suggested its possible involvement in mitochondria-related diseases, such as activation of the ISR by mitochondrial dysfunction in AD patients [91]. However, when the ISR is continuously activated, it may also lead to deleterious effects, such as in excessive ISR activation leading to atherosclerosis [92]. It can be seen that mitochondria also function as signaling organelles, thus participating in maintaining overall cellular homeostasis, and they are closely related to the occurrence and development of diseases [93]. The relationship between these two and their role in MDs still needs to be further validated to help understand mitochondrial pathogenesis and to help understand treatment protocols.

## 7. Treatment of Mitochondrial Diseases

### 7.1. Current Treatments for Mitochondrial Diseases

Due to the complexity of the pathogeneses and clinical manifestations of MDs, there is a lack of effective treatments. In general, symptomatic treatment is more common than etiological treatment [94]. At present, the main treatments can be divided into the following: The first is symptomatic treatment, such as endurance training for patients with hypotonia and hearing aids for patients with hearing impairments. The second is enhancement of the mitochondrial respiratory chain, such as use of the drugs idebenone for Leber’s hereditary optic neuropathy and dichloroacetate for lactic acidosis in MDs [95,96]. The third is increasing mitochondrial biosynthesis with bezafibrate, a PPAR agonist that induces mitochondrial biogenesis by affecting the PPAR-PCG-1α pathway, or resveratrol, which induces PCG-1α activation by activating sirtuins [97,98]. The fourth is gene therapy, in which the phenotypes caused by defective genes are corrected using vectors to introduce normal versions of those genes. At present, major progress has been made in the treatment of Leber’s hereditary optic neuropathy, where 2/5 patients have been reported to show improved vision and the other 3/5 show unchanged vision after treatment [99,100,101]. The fifth includes other treatments, such as the use of antioxidants, cardiolipin protection, the restoration of nitric oxide production, and nucleoside bypass therapy [102,103,104,105].

### 7.2. Potential Therapies for Mitochondrial Diseases Targeting UPR^mt^ and ISR

There is growing evidence that mitochondrial stress responses triggered by gene defects are one of the major factors in MDs, not just OXPHOS defects [106]. With progress in research on the UPR^mt^ and ISR, their roles in maintaining the balance of mitochondrial material metabolism have gradually been emphasized. They also generate some effects consistent with current treatments for MDs: activation of the UPR^mt^ and ISR results in increased synthesis of mitochondria-associated substances, the inhibition of nuclear and mitochondria-encoded gene expression, improved antioxidant responses, the stimulation of mitochondrial function, and enhanced cellular defense mechanisms [107]. There is much evidence that the large accumulation of unfolded, misfolded, or invalid proteins is a typical sign of mitochondria-related neurodegenerative diseases. There is also corresponding experimental evidence that activation of the UPR^mt^ could have a protective effect against these neurodegenerative diseases [108]. Several drugs have been found to improve cellular mitochondrial function and metabolic activity by targeting the UPR^mt^ and ISR. As the main metabolite of curcumin, tetrahydrocurcumin (THC) has stronger antioxidant activity than curcumin. According to a new study in a mouse model, THC activates the UPR^mt^ through the PGC-1α/ATF5 pathway to improve impaired mitochondrial function and reduce ROS production, thereby protecting against pathological cardiac hypertrophy and oxidative stress by antioxidant effects [109]. Choline is a positively charged tetradentate base that is essential for biological metabolism as a component of all biological membranes and a precursor of acetylcholine in cholinergic neurons. Sirtuin 3 (SIRT3) is a mitochondrial deacetylase enzyme that is enriched in metabolically active tissues, such as liver, heart, brain, and brown adipose tissue. SIRT3 is involved in regulating multiple aspects of mitochondrial biology, including ATP production, mitochondrial dynamics, and the UPR^mt^. Choline was shown to regulate metabolic remodeling and the UPR^mt^ through the SIRT3–AMPK pathway to improve mitochondria-associated cardiac hypertrophy in mice [110]. Recent studies have found that pterostilbene, a resveratrol analogue, can also increase the activity of sirtuins through the SIRT/FOXO3a/PGC1α/NRF1 signaling axis to induce UPR^mt^ activation, thereby ameliorating the pathological changes caused by mitochondrial mutations [111,112]. While nicotinamide ribose (NR) is a precursor compound for NAD^+^ and NADH/NAD^+^, its homeostasis is considered to be central to the maintenance of a healthy cellular state. Increasing NAD^+^ levels and restoring NADH/NAD^+^ homeostasis is an attractive therapeutic approach for reversing the development of mitochondrial diseases. Recently, it was shown that NR could also be involved in the maintenance of mitochondrial protein homeostasis in mice with amyotrophic lateral sclerosis (ALS) by activating the UPR^mt^ to alleviate the corresponding neurodegenerative phenotype [113]. All of the abovementioned evidence suggests that the UPR^mt^ may be central to the pharmacological mechanism of a variety of drugs for the treatment of mitochondrial diseases. The use of hypoxia in the treatment of MDs has also been reported, but it is worth exploring whether hypoxia induces the UPR^mt^ and whether the ISR is involved [114]. In addition, activation of the ISR leads to a reduction in overall protein translation, and it was shown that the partial inhibition of protein translation could ameliorate respiratory chain dysfunction [115]. These studies provide strong evidence for the UPR^mt^ and ISR as potential targets for the treatment of MDs. However, it has also been shown that UPR^mt^ and ISR overactivation is detrimental. As previously mentioned, UPR^mt^ overactivation leads to the accumulation of mutant mtDNA. In mice, mutated tRNA synthetase could slowly overactivate the ISR via GCN2, and the resulting axonal peripheral neuropathy leads to the development of Charcot–Marie–Tooth (CMT) disease [116]. Moreover, the regulation of mitochondrial function mediated by the UPR^mt^ and ISR also plays an irreplaceable role in germ cells. For example, it was found in a recent study that oocytes inhibit the activity of respiratory chain complex I by overactivating UPR^mt^, thereby reducing the production of ROS to maintain female fertility [117]. Mitochondria are not just organelles that produce energy for the cell but also play a multifaceted role in maintaining cellular activity. The mitochondrial stress response plays an important role as a bridge between the mitochondria and the cell. More research is required to clarify the multiple roles of the UPR^mt^ and ISR as well as their close association with MDs.

### 7.3. Advantages of UPR^mt^ and ISR in the Treatment of Mitochondrial Diseases

The efficacy of drug therapy for MDs varies due to the type of disease and individual differences, and the side effects of drugs are unavoidable. In addition to drug therapy, gene therapy has shown promising results so far in the treatment of LHON but involves huge medical costs. More human trials are needed to confirm the safety of gene therapy considering that the first patient who received clustered regularly interspaced short palindromic repeats (CRISPR) gene-editing technology died during treatment. In addition, although mitochondrial replacement therapy (MRT) technology for mitochondrial disease prevention significantly reduces the levels of mutated mitochondria, it is faced with controversial ethical issues, and safety and efficacy need to be further studies. In contrast, the UPR^mt^ and ISR have a number of advantages as new targets for the treatment of mitochondrial diseases. Firstly, both the UPR^mt^ and ISR are important in the maintenance of intracellular metabolic homeostasis, so the safety of using them as therapeutic targets is unquestionable. Secondly, the molecular mechanisms of these two reactions have become clearer, and the corresponding signaling pathways have been uncovered in recent years, so the reliability of these pathways as therapeutic targets is indisputable. In addition, the effectiveness of their use as therapeutic targets is obvious because of the significant therapeutic outcomes observed in a large number of animal models of different types of mitochondrial diseases.

## 8. Discussion and Conclusions

In mammals, the ISR and UPR^mt^ are closely associated and play a prominent role in maintaining the homeostasis of mitochondrial components and metabolism. Current studies have also indicated that they both play important roles in the development of mitochondrial diseases. Animal studies have demonstrated that some relevant drugs can ameliorate disease phenotypes by activating the stress response with significant efficacy. Therefore, their potential as new targets for the treatment of MDs is undeniable. However, the current understanding of the exact mechanisms of the ISR and UPR^mt^ and their relationship is still limited for topics such as the activation of corresponding kinases caused by different mitochondrial stresses in the ISR; the downstream regulatory pathways of CHOP, ATF4, and ATF5; and the effect on the UPR^mt^. There are different signaling pathways for activating the UPR^mt^, and it is also unclear whether there is any cross-linking between the different signaling pathways. Moreover, it has been shown that they produce different response outcomes in different cellular environments. The overexpression of UPR^mt^ and ISR also induces impaired mitochondrial function, and the specific regulatory mechanism remains unclear. Further studies are expected to provide evidence to address these questions and to provide more reliable theoretical support for new targeted therapies for mitochondrial diseases.

## Figures and Tables

**Figure 1 cells-12-00020-f001:**
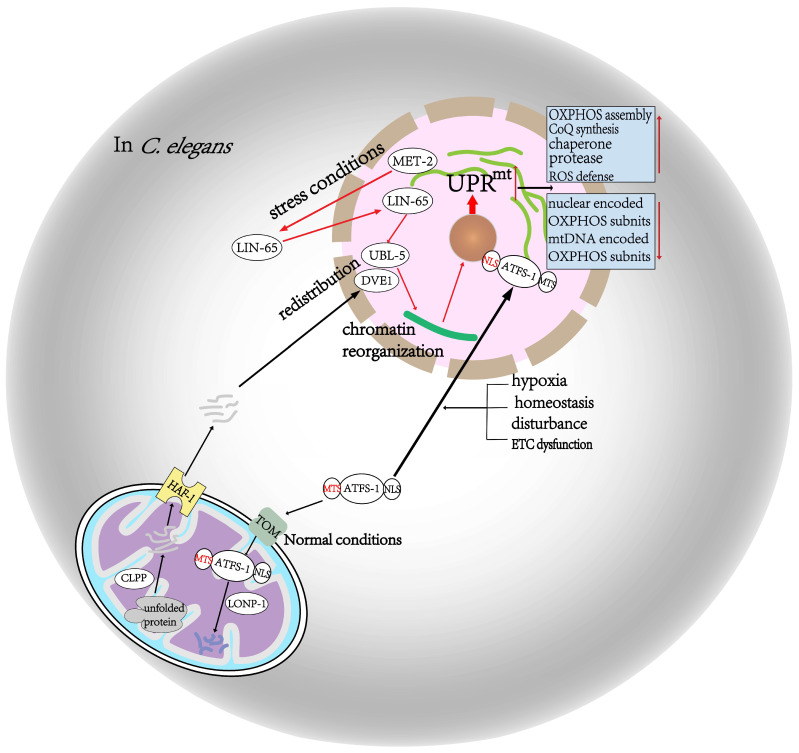
Activation of the mitochondrial unfolded protein response in *C. elegans*. Under normal conditions, ATFS-1 is brought into the mitochondria and undergoes disassembly due to the action of LONP-1. However, when ATFS-1 transport is blocked in cases such as homeostasis disturbance, hypoxia, and ETC dysfunction, it accumulates in the perinuclear region and causes activation of the UPRmt. Unfolded proteins are subsequently broken down into polypeptides by CLPP in the mitochondria and then transported out of the mitochondria by the HAF1 transporter to activate the UBL5 and DEV1 transcriptional complex factors. Under stress situations, UPRmt is activated as a result of UBL5 and DEV1 redistribution and chromatin reorganization, which is the consequence of LIN65 translocation from the cytoplasm to the perinuclear area due to MET2 action. Activation of the UPRmt produces a series of biological changes toward restoring mitochondrial function.

**Figure 2 cells-12-00020-f002:**
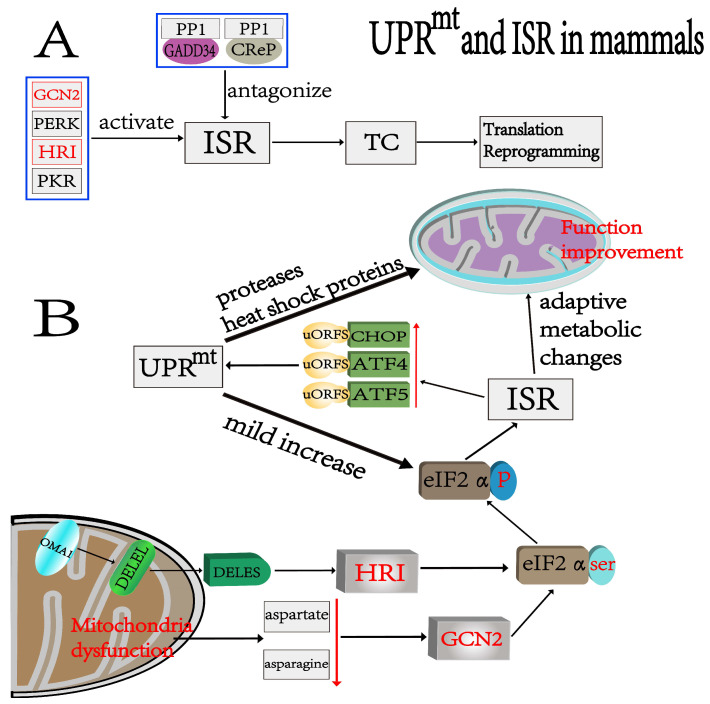
Relationship between UPR^mt^ and ISR and activation process in mammals. (**A**) Four kinases, GCN2, PERK, HRI, and PKR, regulate ISR activation, and two other specialized phosphatases antagonize the ISR. The ISR controls protein translation mainly through the TC. (**B**) Mitochondrial dysfunction induces intracellular stress response mainly through GCN2 and HRI. First, activation of the Oma1 endosomal protein results in conversion of DELEL into DELES. Then, the phosphorylation of eIF-2α serine residues by HRI leads to a series of adaptive metabolic changes that promote the recovery of mitochondrial function and increase the expressions of key UPR^mt^ regulators such as ATF4, ATF5, and CHOP. Second, mitochondrial dysfunction reduces cytoplasmic aspartate and asparagine, which promotes GCN2-induced phosphorylation and, thus, causes corresponding metabolic changes. The activated UPR^mt^ plays a role in maintaining protein homeostasis in mitochondria to enhance mitochondrial function. The UPR^mt^ also causes a mild increase in phosphorylation levels.

**Table 1 cells-12-00020-t001:** Diseases caused by DNA mutation and mutation sites or genes.

Mutation Type	Related Diseases	Mutation Site or Gene
mtDNA mutation		
Point mutation	MELAS syndrome, MERRF LHON Leigh’s syndrome	m.3243A>G m.8344A>G MT-TI, MT-TL1, MTTK, MT-TS1, and MT-TS2 m.11778G>A, m.3460G>A, and m.14484T>C MTFMT, MTTW, MTTV MTTL1, and MTTK
Large deletion	KSS, Pearson, CPEO	mtDNA deletion mtDNA deletion POLG mutation and mtDNA deletion
nDNA mutation		
Respiratory chain subunits	Complex I deficiency Complex II deficiency Complex III deficiency Complex IV deficiency Complex V deficiency	TMEM126B mutation NDUF1V, NDUFV2, NDUFS1, NDUFS2, NDUFA12, NDUFAF2, NDUFAF5, NDUFAF6, FOXRED1, and SDHA SDHAF mutation [82] TTC19 and BCS1L UQCRFS1 mutation [83] SURF1, SCO1, and SCO2 FASTKD2 mutation [84] ATPAF2, TMEM70, and ATP5E
mtDNA replication or transcription	MNGIE MEMSA	TP gene mutation POLG mutation
Protein function	HSP Nerve system	SPG11 mutation SLC25A1 mutation [85]

## Data Availability

Not applicable.

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
