# Peer review of "Mitochondrial Unfolded Protein Response and Integrated Stress Response as Promising Therapeutic Targets for Mitochondrial Diseases"

_cells, 2022, doi:10.3390/cells12010020_

Round 1

Reviewer 1 Report

In this manuscript titled “Mitochondrial unfolded protein response and Integrated Stress Response promising therapeutic target for mitochondrial diseases”, Hedong Lu et al. summarize the role and mechanisms of mitochondrial unfolded protein response (UPRmt) and integrated stress response (ISR). In addition, the authors show the recent process of UPRmt and ISR, and discuss the relationship between them and mitochondrial diseases. The content of this manuscript are interesting, and the manuscript is well written. However, the mechanism of UPRmt and ISR is inadequate (especially in mammals), and need to be supplemented and updated.

Major comments

1. Compared to mitochondrial diseases, the description of UPRmt and ISR is inadequate.

2. The role and mechanism of UPRmt and ISR in mitochondrial diseases needs to be supplemented and fully discussed.

Minor comments

1. “UPRmt” should be “UPRmt”.

2. line 70 “LIN65 islocated in the cytoplasm” should be “LIN65 is located in the cytoplasm”.

3. line 76, “LONP” should be “LONP-1”

Reviewer 2 Report

In this review, the author provides a comprehensive review of the roles and molecular mechanisms of ISR and UPRmt in the pathogenesis of mitochondrial diseases, and discusses their potential applications in the treatment of mitochondrial diseases. In general, this is an interesting and timely topic, but some minor issues need to addressed.

1) Compared to other therapeutics, are there any advantages for UPRmt and ISR in the treatment of mitochondrial diseases. 

2) In Figure 1, the nucleus should be larger than the mitochondria, and some fonts are not very clear, such as chromatin reorganization.

3) In Figure 2, the same issue exists with unclear word.

4) In table 1, diseases caused by DNA mutation and mutation sites or genes should be expanded to include Leigh syndrome at least.

Reviewer 3 Report

In this manuscript the authors review the connection between the mitochondrial unfolded protein response (UPRmt) and the integrated stress response (IRS), and propose that the UPRmt can be a therapeutic target for mitochondrial diseases. Below I write some points:

1. English is poor and needs to be highly improved. The best advice is to ask for the help of an English-speaking person.

2. The review is superficial and it is difficult to agree with one of the main suggestions/conclusions written several times in the review, that is, that UPRmt is a potential therapeutic approach for mitochondrial diseases. At least, it is difficult for me to understand how the change in the activity of the UPRmt pathway could relieve the severe neurologic symptoms of mitochondrial diseases which are due to the inability of mitochondria to synthesize ATP and the larger production of ROS. Could the authors suggest how altering the activity of UPRmt, the synthesis of ATP and the production of ROS by mitochondria will return to normal values, despite the defective function of the respiratory complexes or the ATP synthase?

3. The flawed use of English results in conceptual mistakes or confusion between processes and molecules. For example, when indicated that “mitochondria play an important role in the metabolism of macromolecules such as fatty acid oxidation….” (line 41). The process of fatty acid oxidation occurs in mitochondria but fatty acids are not macromolecules. The sentence is not clear.

4. Another example of conceptual error: “Since the structural proteins of the mitochondrial respiratory chain are co-encoded and translated by nuclear DNA (nDNA) and mitochondrial DNA (mtDNA)” (lines 143-144). Does the word co-encoded give more information than just encoded? Is DNA involved in the translation of these proteins? According to the sentence, yes.

5. To meet the quality required by the journal, the manuscript in its current form needs to be improved for it to be published.

Round 2

Reviewer 1 Report

The revised manuscript quality has been improved, and most of my concerns were addressed and resolved. The manuscript can be accepted now.

Reviewer 3 Report

English was highly improved and the additions and corrections to the manuscript answered most of my questions.